# Clinical and Ultrasound Characteristics of a Difficult-to-Treat Psoriatic Arthritis Population [note 1]

**DOI:** 10.3390/diagnostics15192418

**Published:** 2025-09-23

**Authors:** Georgina Novell, Ana Belén Azuaga, Lucía Alascio, Oriana Omaña, Claudia Arango, Joshua Peñafiel-Sam, Andrés Ponce, Juan Camilo Sarmiento-Monroy, Beatriz Frade-Sosa, José A. Gómez-Puerta, Juan D. Cañete, Julio Ramírez

**Affiliations:** 1Arthritis Unit, Rheumatology Department, Hospital Clínic of Barcelona, 08036 Barcelona, Spain; jnovell@clinic.cat (G.N.); abazuaga@clinic.cat (A.B.A.); lucia.alascio@gmail.com (L.A.); omana@clinic.cat (O.O.); carango@clinic.cat (C.A.); penafiel@clinic.cat (J.P.-S.); aponce@clinic.cat (A.P.); sarmiento@clinic.cat (J.C.S.-M.); frade@clinic.cat (B.F.-S.); jagomez@clinic.cat (J.A.G.-P.); jcanete@clinic.cat (J.D.C.); 2Faculty of Medicine, University of Barcelona, 08007 Barcelona, Spain

**Keywords:** psoriatic arthritis, difficult-to-treat, disease activity, biologic-targeted DMARD failure, paratenonitis

## Abstract

**Background:** Achieving low disease activity or remission in psoriatic arthritis (PsA) remains difficult. The GRAPPA group recently defined difficult-to-treat (D2T) PsA but did not include a time-based criterion. **Objectives:** This study aimed to evaluate the prevalence and features of D2T PsA using several operational definitions. **Methods:** A cross-sectional study at a tertiary center enrolled PsA patients with active disease confirmed by clinical exam and ultrasound. D2T PsA was defined by: (1) failure of ≥1 csDMARD plus ≥2 b/tsDMARDs with different mechanisms of action (GRAPPA definition); (2) ≥2 b/tsDMARDs with different mechanisms of action within 12 months (time-based definition); or (3) failure of >3 b/tsDMARDs with different mechanisms of action (very refractory). Clinical, demographic, radiographic, and ultrasound data were analyzed, and multivariable analyses identified independent associations. **Results:** Seventy-two patients (54.2% female, median age 56, disease duration 84 months) all had active disease (median DAPSA 17); 68% had comorbidities. Enthesitis, dactylitis, and nail disease were seen in 20.8%, 45.8%, and 41.7%. HLA-B27 positivity was 13.9%. Radiographic erosions and ultrasound paratenonitis were present in 37.5% and 33.3%. GRAPPA D2T criteria were met by 23.6%, linked to longer disease duration, higher activity, HLA-B27, comorbidities, and combined therapy. Time-based D2T (12.5%) showed higher DAPSA and nail involvement, with ultrasound paratenonitis and combined therapy independently associated. Very refractory patients (11.1%) only correlated with combined therapy. **Conclusions:** Up to one-fourth of PsA patients remain active despite multiple treatments. D2T PsA is associated with disease duration, comorbidities, activity, HLA-B27, combined therapies. Remarkably, patients who fulfilled the <12-month D2T definition were more likely to present with nail involvement and ultrasound-detected paratenonitis.

## 1. Introduction

Psoriatic arthritis (PsA) is a chronic immune-mediated inflammatory disease that affects both the musculoskeletal system and the skin. It presents with heterogeneous and variable clinical manifestations, including peripheral arthritis, axial involvement, dactylitis, enthesitis, cutaneous and nail psoriasis, and less commonly, extra-articular features such as uveitis and inflammatory bowel disease. PsA is frequently associated with multiple comorbidities, including cardiovascular disease, obesity, metabolic syndrome, and psychosocial conditions such as anxiety and depression. Accurate diagnosis and timely treatment are essential to prevent functional disability and deterioration in quality of life [1,2].

Over the past two decades, advances in understanding the immunopathogenesis of PsA have led to the development of targeted therapies, including biological disease-modifying antirheumatic drugs (bDMARDs) and targeted synthetic DMARDs (tsDMARDs). These agents have significantly improved disease outcomes compared to conventional synthetic DMARDs (csDMARDs). However, data from observational studies and clinical trials indicate that only about 40% of patients treated with biologic therapies achieve minimal disease activity (MDA), underscoring that remission or sustained low disease activity remains an unmet goal for many patients [3,4]. 

Following a two-year process that included a systematic literature review [1], surveys of more than 200 GRAPPA clinicians [5], and input from approximately 600 patients worldwide [6], the Group for Research and Assessment of Psoriasis and Psoriatic Arthritis (GRAPPA) recently proposed consensus-based definitions for complex-to-manage (C2M) and treatment-refractory (TR) PsA. These concepts aim to refine the classification of patients with inadequate treatment responses, recognizing that treatment failure may stem from distinct mechanisms. At the EULAR 2025 Annual Meeting, Proft F. et al. introduced a more structured framework for what has been previously referred to as difficult-to-treat (D2T) PsA [7].

The GRAPPA definitions distinguish between treatment-refractory inflammation and other causes of persistent disease burden. C2M-PsA is defined as “persistent symptoms despite at least one adequate trial of a tsDMARD or bDMARD recommended for PsA,” and includes additional contributing factors such as comorbidities, overlapping syndromes, psychosocial challenges, and treatment-related limitations. In contrast, TR-PsA is defined by failure to respond to three or more therapies with different mechanisms of action (including at least two ts/bDMARDs), in the presence of ongoing symptoms judged problematic by both clinician and patient and supported by objective evidence of inflammation [7]. Although time to treatment failure was acknowledged as clinically relevant, it was ultimately excluded from the formal TR definition—paralleling the approach used in the D2T definition for rheumatoid arthritis [8].

The primary aim of this study is to assess the prevalence and associated characteristics of D2T PsA, applying multiple operational definitions—with and without time to treatment failure as a variable of interest.

## 2. Materials and Methods

### 2.1. Study Design 

This was a cross-sectional, observational, single-center study carried out in a tertiary center (Hospital Clinic, Barcelona, Spain). The principles of the Declaration of Helsinki guided this study, which was approved by the Ethics Committee of the Hospital Clinic of Barcelona (HCB/2022/0212).

### 2.2. Patients

Patients with PsA attending the outpatient clinic underwent sequential collection of clinical and epidemiological information, along with radiographies and ultrasound (US) imaging. To exclude patients not having objective signs of inflammation, only active patients by both clinical examination and US were recruited.

Inclusion Criteria
Adult patients (aged > 18 years) with PsA according to CASPAR criteria [9].Active disease defined by the presence of objective inflammation during the physical assessment and confirmed by US.

Exclusion Criteria
Time of evolution of less than 6 months.Microcrystalline arthritis, defined based on clinical history, laboratory results (serum uric acid, synovial fluid analysis when available), and absence of crystal deposits on imaging.

Three different definitions for difficult to treat PsA were adopted as follows (Figure 1):Failure of ≥1 csDMARD and ≥2 bDMARD or tsDMARD with different mechanism of action (MOA) (GRAPPA definition).Failure of ≥1 csDMARD and ≥2 bDMARD or tsDMARD with different MOAs in <12 months (Time-based definition).Failure of ≥1 csDMARD and ≥3 bDMARD or tsDMARD with different MOAs (Very refractory definition).

Treatment effectiveness was evaluated after a minimum time of three months.

Treatment Failure was defined as the need to discontinue or switch therapy due to persistent active disease (failure to achieve remission or low disease activity as per DAPSA, judged by the treating physician), or due to adverse events.

### 2.3. Procedures

The following procedures were conducted in all patients participating in the study:

### 2.4. Clinical Data Collection

Information was recorded regarding time of evolution, therapies (csDMARDs, glucocorticoids, bDMARD or tsDMARD), and smoking status. The following clinical variables were included: Tender and Swollen joint counts, Disease Activity for PSoriatic Arthritis (DAPSA), enthesitis and dactylitis (past and current), BASDAI for axial PsA patients, skin (Body Surface Area, BSA) and nail disease activity, laboratory markers, Patient Global Assessment (Visual Analog Scale [VAS] 0–10), Physician Global Assessment (VAS 0–10) and Pain Global Assessment (VAS 0–10, completed by the patient).

### 2.5. Radiographic Assesment

Hand and foot X-rays were performed for all patients unless radiographs from the previous year were available. Based on the presence of erosions seen on the X-rays, patients were categorized as either erosive or non-erosive. Joint ankylosis was also evaluated in any joints of the hands or feet. A single reader (JR) assessed the presence of erosive disease (yes/no) and joint ankylosis (yes/no).

### 2.6. Ultrasound Assesment

US evaluations were performed by JR, a rheumatologist with a long experience in US. High-resolution US machine (Canon Aplio a, Canon Medical Systems Corporation, Otawara-shi, Japan) equipped with transducers in the range of 12–18 MHz were used. B-mode and Doppler parameters were adjusted according to the device (pulse repetition frequency range: 400–800 Hz; Doppler frequency: 7–14.1 MHz). US examinations were conducted in both wrists and hands. OMERACT definitions for US elementary lesions were used [10]. Synovial Hypertrophy (SH) and Power Doppler (PD) signals in the wrists and hands (metacarpophalangeal joints 1–5 [MCP]) were scored using a 4-grade semiquantitative scale, according to the definitions provided by Szkudlarek et al. [11]. The assessment of the 1st–5th flexor tendons and the radial flexor tendon was also performed. Paratenon extensor tendon inflammation was defined as a hypoechoic swelling surrounding the extensor digitorum tendon, with or without PD signal [12,13]. Tenosynovitis was defined by tissue within the synovial sheath that is non-displaceable and poorly compressible, visualized in two perpendicular planes, with or without PD signal [10]. Finally, a global US score was calculated for each patient (sum of SH and PD scores for all joints included).

Figure 2 illustrates several examples of pathological findings observed in patients with PsA.

US scans were saved as images for both SH and PD evaluations. These images were assessed independently by two central readers (JR and ABA). Before starting the study, both readers participated in training exercises using static images to score joint inflammation, aiming to establish interobserver agreement. Images representing joints from patients with PsA, covering both active and inactive disease states, were randomly selected and scored simultaneously by each reader. Scoring included semiquantitative scales (0–3) for SH and PD signals, and binary scoring (0–1) for tendinopathy and paratenonitis. Two scoring sessions were conducted between 24 and 72 h apart to evaluate intraclass correlation. Kappa coefficients were interpreted as follows: <0.0 = none, 0.0–0.20 = poor, 0.21–0.40 = modest, 0.41–0.60 = fair, 0.61–0.80 = good, and 0.81–1.00 = excellent.

Intraobserver reliability (kappa coefficients) for JR were 0.63 for SH, 0.87 for PD, 0.95 for tendinopathy, and 0.91 for paratenonitis. For ABA, the corresponding values were 0.68, 0.72, 0.76, and 0.78, respectively. Interobserver reliability yielded kappa coefficients of 0.71 for SH, 0.74 for PD, 0.76 for tendinopathy, and 0.81 for paratenonitis.

### 2.7. Statistical Analysis

Demographic, clinical, and US parameters were examined. Continuous data were expressed as medians with interquartile ranges, whereas categorical data were shown as counts and percentages. Non-parametric methods were employed to compare the distribution of continuous variables across groups, and the Chi-square test was utilized to evaluate differences in categorical variables.

To determine independent relationships, multivariate analyses were performed, controlling for age, sex, disease duration, and all variables with a *p*-value less than 0.1 in the univariate analysis. Statistical significance was defined as a *p*-value of 0.05 or less for all tests.

Kappa statistics were computed to evaluate both intra- and inter-observer agreement. All analyses were conducted using SPSS version 21.0 (SPSS Inc., Chicago, IL, USA).

## 3. Results

### 3.1. Demographic and Clinical Characteristics of the Cohort

A total of 72 patients with PsA were recruited. Of these, 39 were women (54.2%), with a median age of 56 years (IQR 18) and a median disease duration of 84 months (IQR 18). All patients had active disease at the time of assessment, with a median BSA of 1% (IQR 3.25) and a median DAPSA score of 17 (IQR 8). Enthesitis was observed in 15 patients (20.8%), dactylitis in 33 (45.8%), and nail involvement in 30 (41.7%). Ten patients (13.9%) tested positive for HLA-B27.

Radiographically, 27 patients (37.5%) had erosive disease, and 10 (13.9%) had joint ankylosis. US assessment revealed US synovitis (SH > 2 + PD signal) in 23 patients (31.9%), tendon involvement in 35 (49.3%), and MCP paratenonitis in 24 (33.3%).

Comorbidities were present in 49 patients (68%), and 15 (20.8%) had more than three comorbid conditions. Thirty-six patients (50%) were overweight, and 13 (14.4%) were classified as obese. A total of 27 patients (37.5%) reported anxiety or depressive mood disorders.

Regarding prior treatments, 54 patients (75%) had received csDMARDs, 51 (70.8%) had been treated with bDMARDs, and 6 (8.3%) had received tsDMARDs. At the time of the visit, 22 patients (30.6%) were receiving combined therapy with csDMARDs and either bDMARDs or tsDMARDs (Table 1).

### 3.2. Subgroups According to D2T Definitions

#### 3.2.1. D2T GRAPPA Definition

Seventeen patients (23.6%) met D2T Definition 1, defined as failure of ≥1 csDMARD and ≥2 b/tsDMARDs with different MOA. These patients had significantly longer disease duration (*p* = 0.01), a higher number of comorbidities (*p* = 0.045), and higher DAPSA scores (*p* = 0.01). HLA-B27 positivity was more frequent in this group (29% vs. 9%, *p* = 0.049), as was the use of combined therapy (65% vs. 20%, *p* = 0.002) (Table 2). 

In the multivariable analysis, adjusting for age, sex, and disease duration, number of comorbidities (OR 1.9; 95% CI 1–3.5; *p* = 0.032) and combined therapy (OR 19.8; 95% CI 3–1218.5; *p* = 0.002) remained independently associated with this definition.

#### 3.2.2. D2T Time-Based Definition

Nine patients (12.5%) met time-based definition, defined as failure of ≥1 csDMARD and ≥2 b/tsDMARDs with different MOAs within a 12-month period. These patients had significantly higher DAPSA scores (*p* = 0.022), a greater prevalence of nail involvement (23.3% vs. 4.7%, *p* = 0.029), and higher rates of combined therapy use (78% vs. 37%, *p* = 0.002). US-detected MCP paratenonitis was also more frequent in this group (67% vs. 29%, *p* = 0.052) (Table 2). 

In multivariable analysis, after adjusting for age, sex, and disease duration, combined therapy (OR 9.1; 95% CI 1–79.3; *p* = 0.044) and US paratenonitis (OR 23.3; 95% CI 2.2–247.8; *p* = 0.009) were independently associated with this definition.

#### 3.2.3. D2T Very Refractory Definition

Eight patients (11.1%) met this definition, defined as failure of ≥1 csDMARD and ≥3 b/tsDMARDs with different MOAs. This group had significantly more swollen joints (*p* = 0.017), higher DAPSA scores (*p* = 0.052), and a greater frequency of combined therapy use (88% vs. 23%, *p* = 0.002) (Table 2). 

In the multivariate analysis, combined therapy was the only variable independently associated with this definition (OR 24.2; 95% CI 2–283.6; *p* = 0.011).

## 4. Discussion

Despite the wide range of therapeutic options available, many patients with PsA continue to exhibit active disease after multiple lines of advanced therapies [14]. This study highlights the clinical and imaging characteristics associated with D2T PsA—a subgroup of patients that presents distinct challenges in disease management.

The rationale for defining D2T disease lies in identifying a subset of PsA patients with potentially poor long-term prognosis. However, this definition is only meaningful if early biomarkers can be identified and a tailored therapeutic approach implemented. 

We explored the prevalence of D2T disease using three operational definitions: Failure of ≥2 b/tsDMARDs with different MOA, failure of ≥2 MOAs within less than 12 months and failure of ≥3 MOAs. 

Overall, patients meeting any D2T definition had longer disease duration, more comorbidities, and higher disease activity (DAPSA). HLA-B27 positivity and use of combined therapy (csDMARD + b/tsDMARD) were also more frequent in this group. Notably, US-detected MCP paratenonitis and nail involvement were more frequent in patients fulfilling the <12-month D2T definition and may represent a clinical feature associated with rapid treatment changes. These findings suggest that D2T PsA may represent not just treatment resistance but a biologically distinct and more aggressive disease phenotype. 

Managing patients who fail multiple therapies presents several challenges. Many do not respond to conventional or advanced treatments due to comorbidities unrelated to active inflammation. Prior studies have implicated depression, obesity, and metabolic syndrome as contributors to therapeutic failure [15,16,17,18,19,20,21]. We believe it is critical to distinguish “true” D2T patients—where persistent inflammation is the primary issue—from those who are complex to manage due to chronic pain, fatigue, anxiety, or depression [5].

While both groups pose management challenges, they require fundamentally different approaches. To focus on patients with inflammation-driven refractoriness, we included only those with objectively active disease confirmed through clinical and US evaluation. This may explain why comorbidities were only associated with one of the three D2T definitions. 

We incorporated multiple D2T definitions to address the ongoing debate over whether the temporal dimension should be part of the final classification. Although GRAPPA recently published a consensus definition for D2T PsA excluding time as a criterion [7], the expert panel acknowledged that rapid failure of therapies within a short timeframe likely represents a more complex clinical scenario. By including time in one of our definitions, we identified a smaller, more specific subgroup—approximately 10% of active PsA patients—who failed multiple therapies within 12 months. Although our sample size limits strong conclusions, these patients exhibited more active skin disease (higher BSA and significantly more nail involvement) and US-detected paratenonitis, suggesting the idea that rapid refractoriness may signal a more aggressive disease course. 

Previous studies have identified female sex, comorbidities, axial involvement, and skin activity as risk factors for D2T PsA [16,17,18]. In our cohort, the number of comorbidities and HLA-B27 positivity were significantly associated with D2T status under the GRAPPA definition. Nail involvement was particularly elevated in patients meeting the <12-month D2T definition, suggesting that nail disease may contribute to treatment decisions in the short term. As remission is now a central treatment goal in PsA [22], therapeutic decisions in our cohort were likely influenced by disease activity across multiple domains, particularly axial involvement and skin symptoms. 

Combined therapy (csDMARD + b/tsDMARD) was common across all D2T groups, regardless of definition. While this approach is less frequent in PsA than in RA, it remains used by many rheumatologists [23]. Although adding methotrexate to biologics has not consistently shown superior efficacy in PsA, it may enhance drug survival by reducing immunogenicity—particularly with TNF inhibitors [24]. We believe that combined therapy in our cohort reflects both therapeutic practices adapted from RA and deliberate strategies for managing refractory disease. Interestingly, previous studies have not identified combined therapy as a marker for D2T status. 

To our knowledge, this is the first D2T PsA study to incorporate US as part of the patient assessment. While numerical differences in US scores were observed across D2T definitions, statistical significance was only reached when refractoriness was defined as occurring within 12 months. These patients showed significantly higher rates of US-detected MCP paratenonitis—an abnormality more typical of PsA than RA and recently linked to joint ankylosis in our group’s prior work [25]. We propose that both US-detected paratenonitis and nail involvement (enthesis-related lesions) identify a subset of PsA patients at high risk of early therapy failure. However, these markers appear less predictive of treatment changes over the long term.

Our study has some limitations. The cross-sectional design limits causal inference, and the modest sample size reduces statistical power. Moreover, no standardized protocol was applied, and treatment modifications were made at the discretion of the treating clinician. Furthermore, the US scanning protocol used in this study reflects the standard approach employed in our department for evaluating inflammatory arthritis. As it does not include the distal interphalangeal joints or entheses, this may have limited the frequency of US-detected findings in our cohort. Nevertheless, by focusing on a well-defined cohort with objectively confirmed inflammation—excluding patients where chronic pain or psychological comorbidities may bias clinical assessment—we believe our findings provide valuable insights.

Larger, prospective studies are needed to validate our results and determine whether early identification of D2T features can enable more personalized and effective management strategies.

## 5. Conclusions

Up to one-fourth of PsA patients remain active despite multiple therapeutic interventions. D2T PsA was associated with longer disease duration, comorbidities, higher disease activity, HLA-B27 positivity, and use of combined therapy. Nail involvement and US-detected paratenonitis (enthesis-related lesions) were particularly relevant in patients with rapid treatment failure. Identifying these features early may improve risk stratification and inform future management strategies.

## Figures and Tables

**Figure 1 diagnostics-15-02418-f001:**
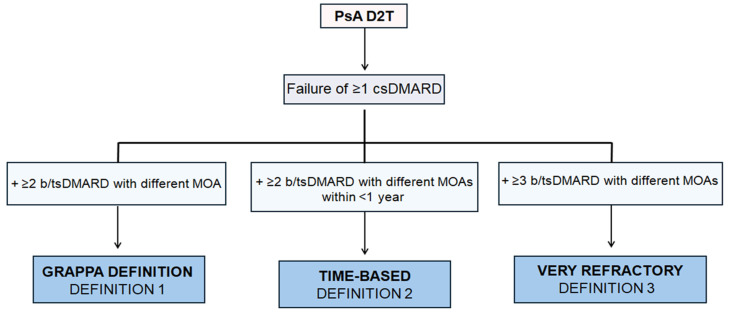
Operational definitions of Difficult-to-Treat Psoriatic Arthritis (D2T PsA). D2T PsA: Difficult-to-Treat Psoriatic Arthritis; csDMARD: Conventional Synthetic Disease-Modifying Antirheumatic Drug; bDMARD: Biologic Disease-Modifying Antirheumatic Drug; tsDMARD: Targeted Synthetic Disease-Modifying Antirheumatic Drug; MOA: Mechanism of Action; GRAPPA: Group for Research and Assessment of Psoriasis and Psoriatic Arthritis.

**Figure 2 diagnostics-15-02418-f002:**
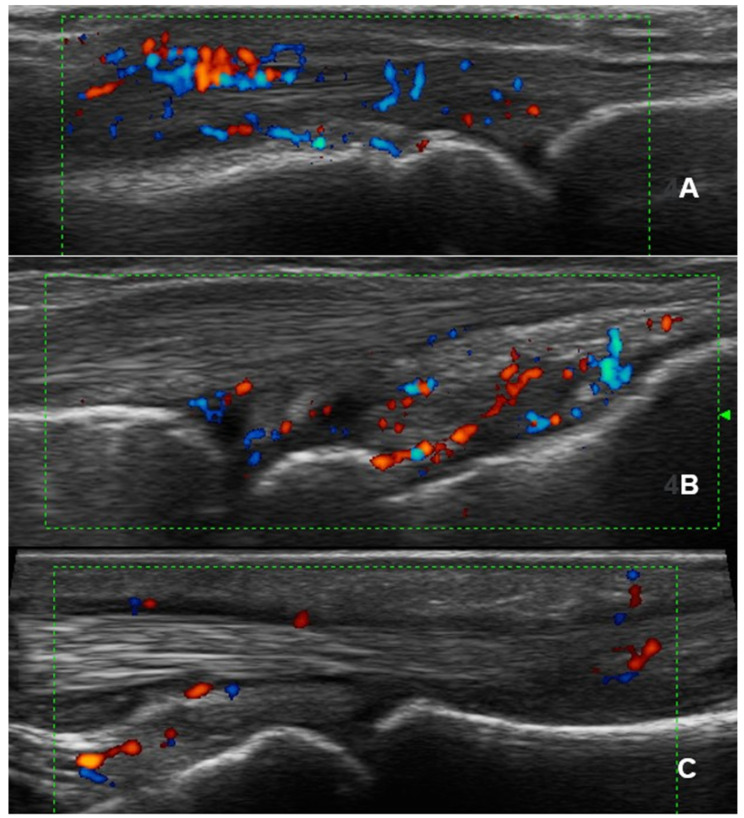
Ultrasound scans of patients with Psoriatic Arthritis. (**A**): Longitudinal scan of the metacarpophalangeal joint demonstrating extensor tendon paratenonitis. (**B**): Longitudinal scan of the carpal joint showing synovial hypertrophy grade II and Power Doppler grade II. (**C**): Longitudinal scan depicting tenosynovitis of the third flexor tendon of the hand.

**Table 1 diagnostics-15-02418-t001:** Clinical and Demographic characteristics of the Cohort (72 patients).

**Age (years), Median (IQR)**	56 (18)
**Female, *n* (%)**	39 (54.2)
**Disease duration (months), Median (IQR)**	84 (18)
**BSA, %, Median (IQR)**	1 (3.25)
**DAPSA, Median (IQR)**	17 (8)
**Enthesitis, *n* (%)**	15 (20.8)
**Dactylitis, *n* (%)**	33 (45.8)
**Nail involvement, *n* (%)**	30 (41.7)
**HLA-B27 positive, *n* (%)**	10 (13.9)
**Erosive disease (X-ray), *n* (%)**	27 (37.5)
**Joint ankylosis (X-ray), *n* (%)**	10 (13.9)
**US Active Synovitis (SH ≥ 2 + PD signal), *n* (%)**	23 (31.9)
**Tendon involvement (US), *n* (%)**	35 (49.3)
**Paratenonitis (US), *n* (%)**	24 (33.3)
**≥1 Comorbidity, *n* (%)**	49 (68)
**≥3 Comorbidities, *n* (%)**	15 (20.8)
**Overweight, *n* (%)**	36 (50)
**Obese, *n* (%)**	13 (14.4)
**Anxiety-depressive mood, *n* (%)**	27 (37.5)
**csDMARD use, *n* (%)**	54 (75)
**bDMARD use, *n* (%)**	51 (70.8)
**tsDMARD use, *n* (%)**	6 (8.3)
**Combined therapy (csDMARD + b/tsDMARD), *n* (%)**	22 (30.6)

bDMARD: biological disease-modifying antirheumatic drug; BSA: Body Surface Area; csDMARD: conventional synthetic disease-modifying antirheumatic drug; DAPSA: Disease Activity for Psoriatic Arthritis; PD: Power Doppler; SH: Synovial Hypertrophy; tsDMARD: targeted synthetic disease-modifying antirheumatic drug; US: Ultrasound.

**Table 2 diagnostics-15-02418-t002:** Clinical and Ultrasound Characteristics of Patients with Difficult-to-Treat Psoriatic Arthritis (D2T PsA) According to Different Criteria.

	GRAPPA Definition (≤2 MOA)		Time-Based Definition (≤2 MOA <12 m)		Very Refractory Definition (≤3 MOA)	
	Yes	No	Yes	No	Yes	No
**Age (years), median (IQR)**	55 (12)	59 (23)	53 (18)	58.5 (20)	56 (24)	58 (19)
**Female, *n* (%)**	8 (47)	31 (56)	6 (67)	33 (52)	4 (50)	35 (45)
**BMI, median (IQR)**	26 (5)	27 (5)	24.5 (7.5)	27 (4.8)	25.5 (7)	27 (5.8)
**Obesity, *n* (%)**	3 (18)	10 (19)	3 (33)	10 (16)	0 (0)	13 (21)
**Cardiovascular risk factors, *n* (%)**	13 (77)	32 (58)	7 (78)	38 (60)	4 (50)	41 (64)
**Fibromyalgia, *n* (%)**	2 (12)	5 (9)	1 (11)	6 (10)	1 (13)	6 (9)
**Anxiety/depressive disorder, *n* (%)**	9 (53)	18 (33)	4 (44)	23 (37)	5 (63)	23 (34)
**Nº Comorbidities, median (IQR)**	3 (3) *	2 (3)	4 (6)	2 (3)	4.5 (4)	2 (3)
**Time of Evolution (m), median (IQR)**	144 (132) *	84 (155)	96 (159)	120 (138)	138 (162)	114 (153)
**DAPSA, median (IQR)**	21 (21) *	16 (7)	26 (23) *	16.5 (7)	26 (21)	17.5 (7)
**BSA (%), median (IQR)**	1 (4)	0 (4)	1.5 (5)	0 (4)	0 (4)	0.5 (4)
**CRP (mg/dL), median (IQR)**	1 (2)	0 (2)	1 (2)	0 (2)	1 (3)	0.5 (2)
**HLAB27, *n* (%)**	5 (29) *	5 (9)	3 (33)	7 (11)	2 (25)	8 (13)
**Dactylitis, *n* (%)**	5 (30)	28 (51)	5 (56)	28 (44)	4 (50)	29 (45)
**Onycopathy, *n* (%)**	10 (59)	20 (36)	7 (78) *	23 (37)	5 (63)	25 (39)
**Erosive Disease, *n* (%)**	8 (47)	19 (35)	4 (44)	23 (37)	5 (63)	22 (34)
**Paratenonitis, US, *n* (%)**	7 (41)	17 (31)	6 (67)	18 (29)	3 (38)	21 (32)
**US Active Synovitis (SH ≥ 2 + PD), *n* (%)**	6 (35)	17 (31)	2 (22)	21 (33)	2 (25)	21 (33)
**Score US, median (IQR)**	8 (14)	6 (5)	10.5 (11)	6 (5)	12.5 (17)	6 (5)
**Combined Therapy, *n* (%)**	11 (65) *	11 (20)	6 (67) *	16 (25)	7 (88) *	15 (23)

BSA: Body Surface Area; CRP: C-Reactive Protein; DAPSA: Disease Activity for Psoriatic Arthritis; m: month; MOA: Mechanism of Action; PD: Power Doppler; SH: Synovial Hypertrophy; US: Ultrasound; * *p* ≤ 0.05.

## Data Availability

The original contributions presented in this study are included in the article. Further inquiries can be directed to the corresponding author.

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
