# Peer review of "Clinical and Ultrasound Characteristics of a Difficult-to-Treat Psoriatic Arthritis Populationâ€"

_diagnostics, 2025, doi:10.3390/diagnostics15192418_

Round 1
Reviewer 1 Report
Comments and Suggestions for Authors
The topic of the manuscript is of interest, due to the high prevalence of difficult-to-treat (D2T) PsA patients in everyday rheumatology practice.
Abstract:
Conclusion – the last sentence should be re-phrased because due to the language barrier the sense of the sentence is changed. In the Methods section the authors define D2T by failure of ≥2 b/tsDMARDs with different MOAs within 12 months (time-based definition).
The conclusion says: ‘Remarkably, nail involvement and US paratenonitis identified D2T patients in less than 12 months.’ The authors probably want to say that D2T PsA patients who had a failure of more than two b/tsDMARDs with different mechanism of action within the 12 months had more nail involvement and US-detected paratenonitis. The way the sentence was written may lead to misinterpretation.
Ultrasound protocol:
Why the sixth extensor compartment of the wrist (the tendon of extensor carpi ulnaris) was included in the US assessment? This tendon is extremely important in RA patients due to its frequent inflammation, both in the establishment of diagnosis of RA and in the assessment of disease activity.
But the current study is about PsA patients. What is the scientific basis for including this tendon in the US evaluation of PsA patients?
Why distal interphalangeal joints were not included in the US assessment? They are one of the most frequently involved in PsA patients.
Discussion section:
‘US-detected MCP paratenonitis and nail disease emerged as potential biomarkers associated with rapid treatment failure.’ Nail disease cannot be considered as a biomarker. The authors should re-phrase.
The following sentence should be re-phrased – ‘Nail involvement was particularly elevated in patients who met the<12-month D2T definition, suggesting that skin activity..’ Nail involvement cannot be considered skin activity. This sentence contains too many inaccuracies.
Why did the authors only include US assessment of the hands and not peripheral entheses so frequently affected in PsA, such as the enthesis of the quadriceps tendon, the proximal and distal patellar enthesis, the enthesis of the Achilles tendon and the plantar fascia?
I suggest that authors include some representative images of the US lesions, for example paratenonitis of the extensor tendon at the level of the MCP joint.
Comments on the Quality of English LanguageIt should be improved.
Author Response
We sincerely thank the reviewers for their constructive feedback and valuable suggestions, which have greatly enhanced the clarity and overall quality of our manuscript. Below, we provide a point-by-point response to each comment.
Additionally, we have made several changes to the Materials and Methods section and improved the language throughout the manuscript to enhance readability and understanding.
Reviewer 1
Abstract
Q1. Conclusion – The last sentence should be re-phrased because due to the language barrier the sense of the sentence is changed. In the Methods section the authors define D2T by failure of ≥2 b/tsDMARDs with different MOAs within 12 months (time-based definition).
The conclusion says: ‘Remarkably, nail involvement and US paratenonitis identified D2T patients in less than 12 months.’ The authors probably want to say that D2T PsA patients who had a failure of more than two b/tsDMARDs with different mechanism of action within the 12 months had more nail involvement and US-detected paratenonitis. The way the sentence was written may lead to misinterpretation
A1: We agree and have re-phrased the sentence in the Conclusions to avoid ambiguity. It now reads:
“Patients who fulfilled the <12-month D2T definition were more likely to present with nail involvement and US-detected paratenonitis.” (Line 32-34)
This wording more accurately reflects our results and avoids the impression that these features define D2T disease in less than 12 months.
Q2. Ultrasound protocol – Why the sixth extensor compartment of the wrist (the tendon of extensor carpi ulnaris) was included in the US assessment? This tendon is extremely important in RA patients due to its frequent inflammation, both in the establishment of diagnosis of RA and in the assessment of disease activity. But the current study is about PsA patients. What is the scientific basis for including this tendon in the US evaluation of PsA patients?
A2: We appreciate the reviewer’s observation. The ultrasound protocol used in this study is the same one we apply in routine clinical practice for evaluating patients with inflammatory arthritis. We acknowledge that certain areas commonly affected in PsA, such as the distal interphalangeal (DIP) joints and entheses, were not included in the protocol. This limitation may have contributed to a lower detection rate of US findings and has now been explicitly mentioned in the Discussion section (line 331-334).
As this is a clinical practice–based protocol, we aimed to limit the number of assessed structures to ensure that the entire US examination could be completed in under 30 minutes. Specifically, the extensor carpi ulnaris (ECU) tendon was included because:
- Although commonly inflamed in rheumatoid arthritis, ECU involvement is not disease-specific and can also occur in PsA, particularly in patients with wrist involvement (Abdelghani KB, et al. J Ultrasound Med. 2023;42:1987–1995).
- Including the ECU tendon enabled us to maintain a comprehensive and standardized scanning approach consistent with our department’s routine protocol for assessing inflammatory arthritis.
Q3. Why distal interphalangeal joints were not included in the US assessment? They are one of the most frequently involved in PsA patients.
A3: We agree that DIP joint involvement is a characteristic feature of PsA. However, DIP ultrasound evaluation was not included in our protocol for two main reasons:
- The reliability of ultrasound assessment at the DIP level is lower compared to MCP, PIP, and wrist joints, primarily due to the small joint size and the frequent presence of degenerative changes in this area.
- Our study specifically focused on lesions with higher interobserver reliability—such as MCP paratenonitis, synovitis, and flexor tendon involvement—which were central to our study hypothesis.
Q4. Discussion section-‘US-detected MCP paratenonitis and nail disease emerged as potential biomarkers associated with rapid treatment failure.’ Nail disease cannot be considered as a biomarker. The authors should re-phrase.
The following sentence should be re-phrased – ‘Nail involvement was particularly elevated in patients who met the<12-month D2T definition, suggesting that skin activity..’ Nail involvement cannot be considered skin activity. This sentence contains too many inaccuracies.
A4. We fully agree. We have modified both sentences in the text:
“Nail involvement was more frequent in patients fulfilling the <12-month D2T definition and may represent a clinical feature associated with rapid treatment changes.” (Line 275-277)
We now avoid using the term “biomarker” in this context.
“Nail involvement was particularly elevated in patients meeting the <12-month D2T definition, suggesting that nail disease may contribute to treatment decisions in the short term.” (Line 306-308)
This avoids equating nail disease with skin activity.
Q6. Why did the authors only include US assessment of the hands and not peripheral entheses so frequently affected in PsA, such as the enthesis of the quadriceps tendon, the proximal and distal patellar enthesis, the enthesis of the Achilles tendon and the plantar fascia?
A6: We acknowledge this limitation and agree that enthesis involvement is highly relevant in PsA. In this study, we applied our department’s standard ultrasound protocol for inflammatory arthritis, which focuses on hand and wrist structures. This approach was chosen to ensure feasibility within the clinical visit and to maintain high interobserver reliability. We have now included this clarification in the Limitations section. (Line 331-334)
Q7. Suggestion to include representative US images. I suggest that authors include some representative images of the US lesions, for example paratenonitis of the extensor tendon at the level of the MCP joint.
A7: We thank the reviewer for this helpful suggestion. Representative images of MCP paratenonitis, flexor tenosynovitis, and synovitis have now been included in Figure 2.
Reviewer 2 Report
Comments and Suggestions for Authors
Clinical and Ultrasound Characteristics of a Difficult-to-Treat Psoriatic Arthritis Population
Overall the manuscript evaluated the prevalence and features of difficult to treat psoriatic arthritis based on some clinical and ultrasound characteristics. Because this was an observational study so that some variables needed to be defined more clearly. There were some comments for the manuscript:
1. The authors should not use many abbreviations in the abstract if it is not necessary because this could confuse the reader.
2. Regarding materials and methods: please help to clarify how to diagnose patients with microcrystalline arthritis. Moreover, most of the patients will experience treatment prior so how can the author control the bias in this study?
3. Definition of difficult to treat PsA lack of time to evaluate the efficacy after initiating treatment
4. Definition of failure needs to be explained in more details: improvement less than how many percent will be defined as failure.
5. Line 105: correct the mistake /2Failure
Author Response
We sincerely thank the reviewers for their constructive feedback and valuable suggestions, which have greatly enhanced the clarity and overall quality of our manuscript. Below, we provide a point-by-point response to each comment.
Additionally, we have made several changes to the Materials and Methods section and improved the language throughout the manuscript to enhance readability and understanding.
Reviewer 2
Overall the manuscript evaluated the prevalence and features of difficult to treat psoriatic arthritis based on some clinical and ultrasound characteristics. Because this was an observational study so that some variables needed to be defined more clearly. There were some comments for the manuscript:
Q1. The authors should not use many abbreviations in the abstract if it is not necessary because this could confuse the reader.
A1: We agree and have reduced the use of abbreviations in the Abstract, retaining only those that are well-known and strictly necessary.
Q2. Regarding materials and methods: please help to clarify how to diagnose patients with microcrystalline arthritis.
A2: Patients with microcrystalline arthritis were excluded based on clinical history, laboratory results (serum uric acid, synovial fluid analysis when available), and absence of crystal deposits on imaging. This has been added to the Exclusion Criteria section. (Line 100-102)
Q3. Moreover, most of the patients will experience treatment prior so how can the author control the bias in this study?
A3: We acknowledge the potential for bias inherent to observational studies in which no predefined treatment protocol was followed and therapeutic decisions were made at the discretion of the treating clinician. To minimize this bias, we included only patients with objectively active disease, confirmed by both clinical assessment and ultrasound examination. This approach was intended to reduce the risk of misclassification due to residual pain or non-inflammatory causes. We have now clarified this point in the Discussion section as a study limitation. (Line 329-331)
Q4 Definition of difficult to treat PsA lack of time to evaluate the efficacy after initiating treatment
A4: We appreciate this observation. We consider three months to be the minimum time required to evaluate the efficacy of biological or JAK inhibitor therapies. This clarification has now been added to the text. (Line 111)
Q5. Definition of failure needs to be explained in more details: improvement less than how many percent will be defined as failure.
A5: We have now clarified in the Methods:
“Treatment failure was defined as the need to discontinue or switch therapy due to persistent active disease (no achievement of remission or low disease activity as per DAPSA, judged by the treating physician), or due to adverse events.” (Line 112-114)
This definition is now explicitly stated.
Q6. Line 105: correct the mistake /2Failure
A6: Corrected.
Round 2
Reviewer 1 Report
Comments and Suggestions for Authors
The authors have revised their manuscript as per the reviewer's recommendations.